

# The effect of running versus cycling high-intensity intermittent exercise on local tissue oxygenation and perceived enjoyment in 18–30-year-old sedentary men

Yuri Kriel, Christopher D. Askew and Colin Solomon

School of Health and Sports Sciences, University of the Sunshine Coast, Sippy Downs, QLD, Australia

## ABSTRACT

**Background:** High-intensity interval training (HIIT) has been proposed as a time-efficient exercise format to improve exercise adherence, thereby targeting the chronic disease burden associated with sedentary behaviour. Exercise mode (cycling, running), if self-selected, will likely affect the physiological and enjoyment responses to HIIT in sedentary individuals. Differences in physiological and enjoyment responses, associated with the mode of exercise, could potentially influence the uptake and continued adherence to HIIT. It was hypothesised that in young sedentary men, local and systemic oxygen utilisation and enjoyment would be higher during a session of running HIIT, compared to a session of cycling HIIT.

**Methods:** A total of 12 sedentary men (mean ± SD; age 24 ± 3 years) completed three exercise sessions: a maximal incremental exercise test on a treadmill (MAX) followed by two experiment conditions, (1) free-paced cycling HIIT on a bicycle ergometer (HIITCYC) and (2) constant-paced running HIIT on a treadmill ergometer (HIITRUN). Deoxygenated haemoglobin (HHb) in the gastrocnemius (GN), the left vastus lateralis (LVL) and the right vastus lateralis (RVL) muscles, oxygen consumption ($VO_2$), heart rate (HR), ratings of perceived exertion (RPE) and physical activity enjoyment (PACES) were measured during HIITCYC and HIITRUN.

**Results:** There was a higher HHb in the LVL ($p = 0.001$) and RVL ($p = 0.002$) sites and a higher $VO_2$ ($p = 0.017$) and HR ($p < 0.001$) during HIITCYC, compared to HIITRUN. RPE was higher ($p < 0.001$) and PACES lower ($p = 0.032$) during HIITCYC compared to HIITRUN.

**Discussion:** In sedentary individuals, free-paced cycling HIIT produces higher levels of physiological stress when compared to constant-paced running HIIT. Participants perceived running HIIT to be more enjoyable than cycling HIIT. These findings have implications for selection of mode of HIIT for physical stress, exercise enjoyment and compliance.

Corresponding author
Yuri Kriel, ykriel@usc.edu.au

## INTRODUCTION

High-intensity interval training (HIIT) is broadly defined as repeated bouts of short to moderate duration exercise (10 s to 4 min) completed at a relatively high intensity, separated by periods of low-intensity exercise or passive rest (30 s to 5 min) (*Billat, 2001*; *Buchheit & Laursen, 2013a*, *2013b*; *Laursen & Jenkins, 2002*). HIIT of a short duration has been proposed as a time-efficient exercise format to improve exercise adherence, thereby helping to address the chronic disease burden associated with sedentary behaviour (*Gillen & Gibala, 2014*). However, there is a relative and absolute scarcity of clinical exercise professionals to provide formalised exercise guidance to sedentary individuals (*Cheema, Robergs & Askew, 2014*) looking to initiate HIIT. Given the extensive coverage of the beneficial health effects of HIIT in previous literature (*Batacan et al., 2017*; *Kessler, Sisson & Short, 2012*) and 'go as hard as you can' formats of HIIT via media outlets, it seems likely that a proportion of sedentary individuals looking to increase their physical activity levels will attempt HIIT in recreational settings unsupervised, self-selecting the modes of exercise (running and cycling) available to them.

Running elicits a greater cardiorespiratory response ($VO_2$ and heart rate (HR)) than cycling during incremental and submaximal exercise, at matched relative and absolute workloads above and below the anaerobic threshold (*Abrantes et al., 2012*; *Scott et al., 2006*). However, when selecting running versus cycling to perform a session of HIIT, workloads are inherently different. This difference is due, in part, to how running and cycling ergometers are utilised to induce the requisite physiological stress inherent in the HIIT bouts and protocols (*Ben Abderrahman et al., 2013*; *Weston et al., 2014*). It is not known whether running elicits a greater cardiorespiratory response than cycling, in sedentary individuals, during short-duration HIIT protocols. Additionally, while acute physiological responses linked to positive health and performance benefits are induced by bouts of either running and cycling HIIT (*Kessler, Sisson & Short, 2012*; *Logan et al., 2014*; *Molmen-Hansen et al., 2012*; *Weston, Wisloff & Coombes, 2014*; *Whyte et al., 2013*; *Whyte, Gill & Cathcart, 2010*), there is no scientific literature comparing the effects of these exercise modes on physiological responses during HIIT in the same cohort of sedentary individuals (*Buchheit & Laursen, 2013a*). The physiological responses, and therefore benefits, elicited by running and cycling HIIT, as conducted in recreational exercise settings by unsupervised sedentary individuals, are expected to differ due to differences in the absolute workload able to be achieved (*Abrantes et al., 2012*), cardiorespiratory responses (*Abrantes et al., 2012*; *Hill, Halcomb & Stevens, 2003*; *Scott et al., 2006*), muscle activation (*Bijker, de Groot & Hollander, 2002*) and systemic oxygen utilisation (*Carter et al., 2000*) between these two modes of exercise. Therefore, it is necessary to determine which mode will potentially provide the largest physiological perturbation and acute training response in unsupervised exercise scenarios. Comparing the physiological responses during each bout of the cycling and running HIIT protocols will allow a more detailed examination of potential differences than analysing these variables at the protocol level.

A benefit of HIIT exercise, increased cardiorespiratory fitness, has been attributed partly to increases in mitochondrial content and function (*Jacobs et al., 2013*; *Russell et al., 2014*). It is plausible that the increase in oxygen utilisation at the local tissue level during acute bouts of HIIT contributes to the stimulus for these adaptations. Near-infrared spectroscopy (NIRS) is a non-invasive method for the measurement of the change in concentration of oxyhaemoglobin ($\Delta O_2Hb$) (oxygen availability) and deoxyhaemoglobin ($\Delta HHb$) (oxygen utilisation) at the local tissue level. Oxygen utilisation, measured via NIRS, has been described during running HIIT in the quadriceps and hamstring (*Buchheit, Hader & Mendez-Villanueva, 2012*; *Buchheit & Ufland, 2011*) and during cycling HIIT in the quadriceps (*Chin et al., 2011*; *Koga et al., 2007*). The HHb in the locomotor muscles is increased during cycling and running HIIT bouts, compared to pre-exercise values (*Buchheit et al., 2009*; *Dupont et al., 2007*), but has routinely been assessed at only a single measurement site (*Perrey & Ferrari, 2018*). During HIIT, differences in the oxygen utilisation response may exist between locomotor muscles, as a function of exercise mode, due to differences in muscle composition (*Hagström-Toft et al., 2002*; *Houmard et al., 1998*) and muscle activation (*Bouillon et al., 2016*). It is probable that, similar to systemic measures of oxygenation in other formats of exercise (*Abrantes et al., 2012*; *Scott et al., 2006*), running HIIT will elicit a greater local oxygen utilisation response than cycling HIIT at each measurement site. However, this is yet to be demonstrated. Investigation of the multi-site local oxygen utilisation responses during repeated bouts of HIIT in sedentary individuals will provide specific information as to which mode causes the largest increase in oxygen utilisation, a potential stimulus for improved mitochondrial function in muscle fibres, and therefore an important consideration when evaluating HIIT in a health-related context.

HIIT is often associated with discomfort (*Astorino et al., 2012*; *Oliveira et al., 2013*). Aversive exercise experiences have a direct effect on perception of exertion, exercise enjoyment and therefore adherence (*Kwan & Bryan, 2010*; *Williams et al., 2008*) in sedentary individuals, a population which often has low intrinsic motivation to exercise (*Aaltonen et al., 2014*). It is necessary to use perceptual data to determine the factors that may facilitate or impede the commencement or continuation of HIIT, if HIIT is to be an effective preventative health initiative (*Coyle, 2005*; *Kessler, Sisson & Short, 2012*; *Logan et al., 2014*; *Whyte, Gill & Cathcart, 2010*). Perceived exercise intensity affects sedentary individuals' exercise enjoyment levels and ongoing compliance (*Jung, Bourne & Little, 2014*), with lower intensity exercise associated with higher levels of enjoyment (*Williams et al., 2008*) and improved adherence rates in novice exercisers (*Pescatello & American College of Sports Medicine, 2013*). However, this inverse relationship is not always strong (*Rhodes, Warburton & Murray, 2009*) and is complicated by various factors, including exercise mode. At a fixed submaximal intensity (*Thomas et al., 1995*) and during incremental exercise (*Abrantes et al., 2012*), cycling elicits a higher perception of effort than running. If, during bouts of HIIT likely to be adopted by sedentary individuals, cycling also elicits a higher perception of effort than running (and hence a lower level of enjoyment) exercise mode potentially has implications for the practical uptake of HIIT in a sedentary population.

It was hypothesised that in young sedentary individuals $\Delta$[HHb], $VO_2$, HR and the Physical Activity Enjoyment Scale (PACES) would be higher and ratings of perceived exertion (RPE) lower during running HIIT bouts, compared to cycling HIIT bouts.

## METHODS

### Ethics statement

This research study was approved by the human research ethics committee of the University of the Sunshine Coast (S/13/472). All participants received a research study information sheet before providing written informed consent.

### Experiment design

The study consisted of three testing sessions: a maximal incremental exercise test conducted on a treadmill ergometer (MAX) followed by two experiment conditions: (1) a protocol of free-paced HIIT conducted on a bicycle ergometer (HIITCYC); and (2) a protocol of constant-paced HIIT conducted on a treadmill ergometer (HIITRUN). The HIIT conditions were randomized to control for any possible order effect. All testing sessions were separated by three to seven days to minimise the influence of any potential carry-over effects between testing sessions. The $\Delta$HHb in the gastrocnemius (GN), the left vastus lateralis (LVL) and the right vastus lateralis (RVL) muscles, $VO_2$, HR, RPE and the PACES were measured during the HIIT sessions. The format of the HIIT testing sessions and the timing of measurements are illustrated in Fig. 1.

### Participants

The participant group consisted of 12 men who met the inclusion criteria of being aged 18–30 years; currently completing less than 150 min of moderate intensity or 75 min of vigorous intensity activity per week; reporting no cardiovascular and metabolic disease; taking no medications; having no known health-related issues that would be made worse by, or inhibit participation in the study. Descriptive physical characteristics of participants are presented in Table 1.

## PROCEDURES AND EQUIPMENT

### Screening procedures

At the initial session, participants completed risk screening questionnaires, a physical activity log and physical characteristics of height, mass, resting pulmonary function and adipose tissue thickness (ATT) were measured (Table 1), as previously described (*Kriel et al., 2016*). Briefly, the physical activity log was used to ensure that participants' activity levels over the previous three months met the definition of sedentary for the purposes of this study (i.e., not achieving the current minimal recommendations for exercise participation to obtain health benefits) (*Barnes et al., 2012*). Participants were asked to not perform exercise in the 24 h prior to each session and to not consume any caffeine, alcohol or a large meal in the 4 h preceding each session (*Goldstein et al., 2010*). It was confirmed at each testing

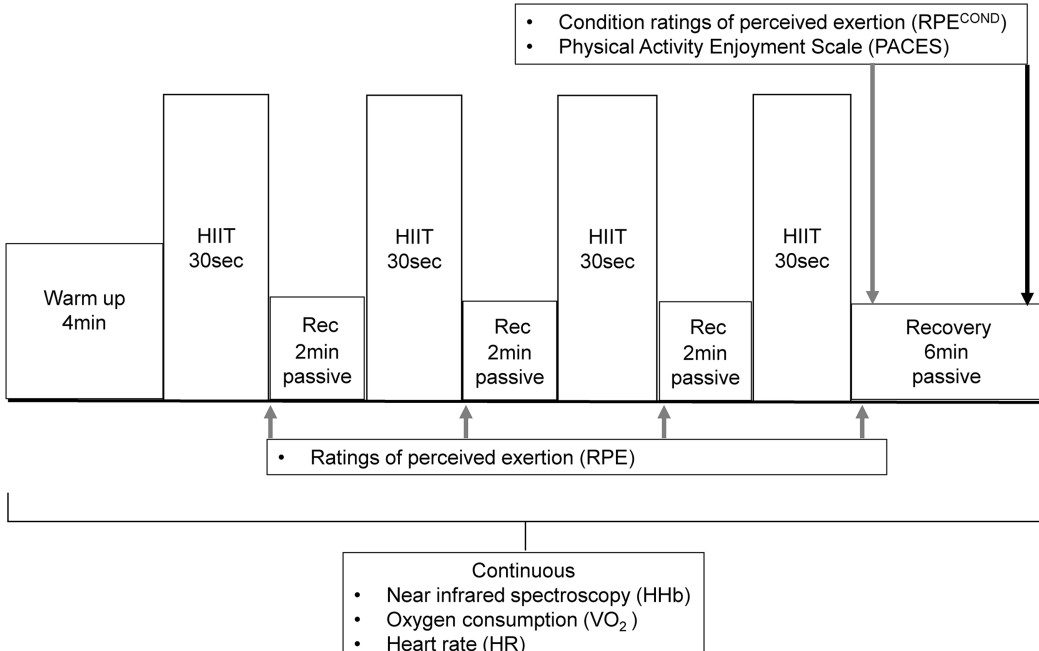

**Figure 1 The format and timing of measurements of the HIIT sessions.**

| Table 1 Participant characteristics. | |
|---|---|
| Height (cm) | 176 ± 7.07 |
| Weight (kg) | 78.9 ± 13.9 |
| Age (years) | 24 ± 3 |
| $VO_{2peak}$ during MAX (ml · kg · min$^{-1}$) | 43.5 ± 4.30 |
| Heart rate max (bpm) | 189 ± 9 |
| Peak treadmill running speed during MAX (km · h$^{-1}$) | 12.2 ± 0.94 |
| Peak RPE$^{COND}$ value during MAX | 8 ± 1 |
| Left vastus lateralis skinfold (mm) | 10.4 ± 4.48 |
| Right vastus lateralis skinfold (mm) | 11.0 ± 4.97 |
| Gastrocnemius skinfold (mm) | 10.9 ± 3.87 |
| FVC (L) | 5.46 ± 0.75 |
| FVC % pred (%) | 106 ± 11 |
| FEV$_1$ (L) | 4.56 ± 0.80 |
| FEV$_1$ % pred (%) | 105 ± 16 |

Notes:
Data are (mean ± SD) ($n = 12$).
$VO_{2peak}$, Peak oxygen uptake; RPE$^{COND}$, Rating of perceived exertion for each condition; FVC, Forced vital capacity; FEV$_1$, Forced expiratory volume in 1 s.

session that participants were appropriately fed and hydrated, in line with existing pre-exercise nutrition and hydration guidelines (*Convertino et al., 1996*; *Garzon & Mohr, 2014*).

## Exercise sessions

Prior to the initial exercise session, participants were familiarised with the testing protocols, the cycle ergometer (Veletron; Racermate, Seattle, WA, USA) and the treadmill (T200; Cosmed, Rome, Italy). A safety harness was worn during all treadmill sessions.

Each exercise session began with a 3 min baseline data collection period during which the participant either sat on the cycle ergometer or stood on the treadmill. The baseline period was followed by a 4 min warm up period, consisting of either cycling against a constant resistance of 60 Watts (W) (HIITCYC) or walking at a 10% gradient and an individually determined speed calculated to produce approximately 60 W of resistance (HIITRUN), calculated from the rate of vertical displacement (running speed and gradient) and body mass.

During the maximal incremental exercise test, the warm up was followed by an incremental speed protocol at a fixed 10% gradient, with an initial running speed of $8 \, \text{km} \cdot \text{h}^{-1}$, increasing by $1 \, \text{km} \cdot \text{h}^{-1}$ every 30 s until volitional cessation. The peak treadmill running speed achieved during MAX was used as the participants' running speed during the HIITRUN bouts. The 10% grade was chosen to simulate individuals performing HIIT by running up a steep incline, to enable mechanical power to be calculated and to provide an intensity of exercise that would produce a maximal sprint effort without relying primarily on running speed. It was necessary to determine a realistic maximal running speed for the 30 s HIITRUN bouts, which would potentially not have been achieved with longer stage durations in untrained sedentary participants during the maximal incremental test, due to fatigue-inducing exercise cessation prior to the attainment of maximal running speed (*Kang et al., 2001*; *Kirkeberg et al., 2011*).

The format of the HIIT experiment conditions, as illustrated in Fig. 1, have been described previously (*Kriel et al., 2016*). Briefly, the warm up was followed by four 30 s bouts of HIIT, with 2 min passive recovery periods separating each of the HIIT bouts. The four bouts of HIIT during HIITCYC consisted of repeat Wingate sprints. The four bouts of HIIT during HIITRUN consisted of participants running at the maximal speed achieved during the maximal incremental test (MAX), for each of the four bouts. During the passive recovery periods, participants were instructed to sit (HIITCYC) or stand (HIITRUN) as still as possible to reduce movement artefact in the NIRS data. Following the final bout, there was a 6 min passive recovery period. The time-efficient HIIT protocol format was based upon protocols used in trained and untrained populations (*Buchheit et al., 2010*, *2012*; *Burgomaster et al., 2005*, *2008*; *Dupont et al., 2007*; *Gibala et al., 2006*; *Gibala & McGee, 2008*).

## Tissue oxygenation

Changes in local tissue oxygenation were measured continuously, as illustrated in Fig. 1, and as described previously (*Kriel et al., 2016*). Briefly, the changes in the relative concentration of HHb ($\Delta$[HHb]) were measured using a NIRS system (three x PortaMon devices; Artinis Medical Systems BV, Zetten, The Netherlands). The PortaMon devices were placed on the skin directly over the muscle belly of three locomotor muscles: the LVL, the RVL and the left GN. Test-retest reliability of the HHb data from the NIRS system

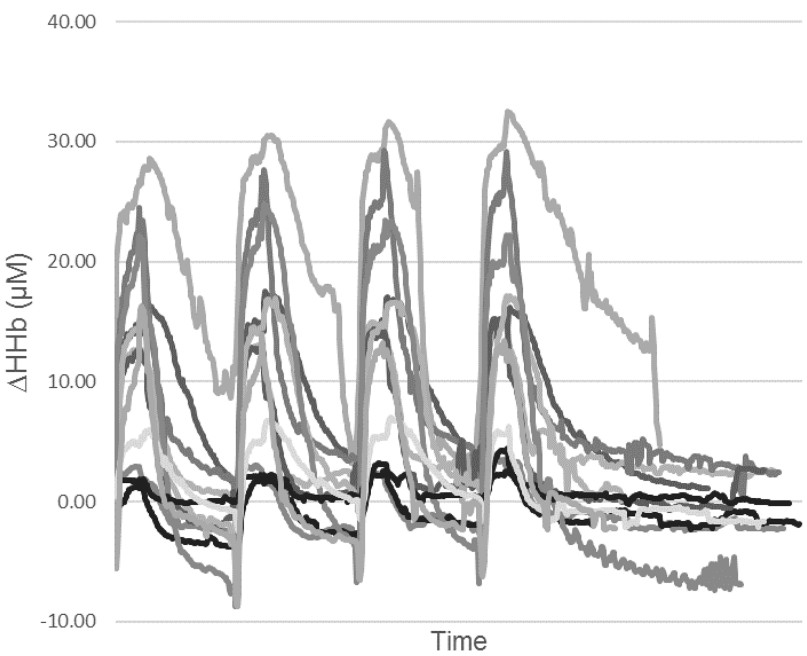

**Figure 2 Individual relative change from baseline of deoxygenated haemoglobin (HHb) for the RVL during the HIITCYC condition (*n* = 12).**   

was examined prior to this study, providing acceptable absolute reliability values of the baseline data for each site (Typical Error: VL = 0.4 μM, GN = 0.8 μM) (*Kriel et al., 2016*). Furthermore, previous research has shown that the NIRS method provides acceptable reliability for the measurement of HHb in active muscle tissue (*Austin et al., 2005*; *Muthalib et al., 2010*). For all testing, the same device was used at the same measurement site. To ensure placement consistency, positioning of the individually labelled NIRS devices was referenced to anatomical landmarks as detailed previously (*Buchheit et al., 2009*, *2011*; *Dupont et al., 2007*; *Prieur & Mucci, 2013*; *Smith & Billaut, 2010*). The location of devices was marked with an ink pen at the first session and participants were required to maintain the marks between subsequent sessions.

For this study, only $\Delta[HHb]$ values are presented, as discussed previously (*Kriel et al., 2016*). Briefly, the $\Delta[HHb]$ data are potentially unaffected by changes in perfusion, blood volume or arterial haemoglobin concentration (*Adami et al., 2015*; *Jones et al., 2009*; *Wang et al., 2006*). Movement artefact was present in NIRS data collected at the LVL, RVL and GN sites during the passive recovery periods of the HIIT sessions, attributed to a low signal to noise ratio. Therefore, recovery period data were not included in further analysis. Similar technical difficulties in NIRS data collection, leading to data exclusion, have been reported (*Mekari et al., 2015*). The inter-individual variability in the $\Delta[HHb]$ data (range −5.77 to 32.33 μM during the HIIT bouts) is presented in Fig. 2 for the RVL during HIITCYC.

## Systemic oxygen consumption, HR and mechanical power

Systemic oxygen consumption ($VO_2$) data were collected continuously, as illustrated in Fig. 1, using a respiratory gas analysis open circuit spirometry system (Parvo Medics, Sandy, UT, USA). Standardised calibration and methods were used (*Macfarlane & Wu,*
*2013*). A HR monitor (RS400; Polar Electro, Kempele, Finland) was used to measure and record HR, as illustrated in Fig. 1. A crank-based power meter (SRM Science; Schoberer Rad Meßtechnik, Julich, Germany) was used during HIITCYC to measure and record mechanical power output. The power output during the HIITRUN bouts was a product of the constant speed, the fixed 10% gradient and the participant's body weight using the equations: Power = Work/Time; Work = body weight × total vertical distance (speed × gradient × time), used previously when comparing high-intensity bouts of uphill running to non-steady state cycling (*Scott et al., 2006*).

### Ratings of perceived exertion

To determine perceived exertion, participants were provided with a standardised description of the CR10 RPE scale (*Borg, 1982*) and the scale's purpose, including memory anchoring of the scale (an explanation of the sensations associated with the high and low scale categories) (*Gearhart et al., 2004*). Participants were asked to provide an RPE score immediately after each bout of HIIT as well as give a 'Condition' RPE score (RPE$^{COND}$) 1 min after the final (fourth) bout. This RPE$^{COND}$ score was used to compare the overall perception of exertion between the HIIT conditions.

### Physical activity enjoyment scale

To determine enjoyment levels in response to each condition, participants completed the PACES questionnaire within 5 min of completing each condition. The PACES was used to compare enjoyment levels between the HIIT conditions. The PACES consists of eighteen items on a seven-point bipolar scale. A minimum total score of 18 and a maximal total score of 126 is possible. The PACES is a reliable and valid measure of enjoyment during HIIT (*Oliveira et al., 2013*; *Tritter et al., 2013*).

### Data calculation

All NIRS data were collected at 10 Hz, smoothed using a 10-point moving average and then averaged to 1 s periods for statistical analysis. The NIRS data were expressed as units of change (μM) from the mean value of the 30 s of baseline data preceding the start of exercise ($\Delta$[HHb]). VO$_2$ data were averaged over 5 s periods while HR and HIITCYC power data were averaged at 1 s intervals initially. The NIRS, HR, VO$_2$ and HIITCYC power data were then time aligned. The time periods of data corresponding to the four 30 s bouts of HIIT were then identified. Mean 30 s values were then calculated for all dependant variables for each bout of HIIT, providing a single value per bout for statistical analysis. RPE and PACES data provided a single value per measurement time for statistical analysis.

### Statistics

Statistical tests were performed using the IBM SPSS Statistics (version 22; IBM Corporation, Armonk, NY, USA) program. Data were initially tested for normality of distribution using the Shapiro–Wilk test. A three factor, repeated-measures analysis of variance (ANOVA) was used to analyse the effect of condition, bout and site on the dependant variable: $\Delta$[HHb]. A two factor, repeated-measures ANOVA was used to analyse the effect of condition and bout on the dependant variables of VO$_2$, HR,

mechanical power and RPE. If a significant main effect or interaction effect was identified, a Bonferroni's post hoc test was used to make pair wise comparisons. A paired samples *T*-test was used to analyse the effect of condition on PACES scores. All variables are presented as mean ± SD. For all statistical analyses, a *p* value of < 0.05 was used as the level of significance. Effect size estimates are indicated using partial $\eta^2$ values.

## RESULTS

### Tissue oxygenation

For the mean $\Delta$[HHb] for all sites, conditions and bouts combined, there was a main effect for site ($p = 0.002$, $F = 6.057$, $\eta_p^2 = 0.377$), condition ($p < 0.001$, $F = 43.392$, $\eta_p^2 = 0.813$) and bout ($p = 0.022$, $F = 5.641$, $\eta_p^2 = 0.361$). There were condition $\times$ site ($p < 0.001$, $F = 8.031$, $\eta_p^2 = 0.445$), condition $\times$ bout ($p = 0.002$, $F = 9.643$, $\eta_p^2 = 0.491$) and site $\times$ bout ($p < 0.001$, $F = 7.615$, $\eta_p^2 = 0.432$) interactions. For the mean $\Delta$[HHb] for each condition, there was a main effect for site for HIITCYC ($p = 0.001$, $F = 6.846$, $\eta_p^2 = 0.384$) and HIITRUN [($p < 0.001$, $F = 8.981$, $\eta_p^2 = 0.473$). There was a main effect for bout ($p < 0.001$, $F = 15.849$, $\eta_p^2 = 0.590$) and site $\times$ bout interactions ($p = 0.003$, $F = 5.353$, $\eta_p^2 = 0.327$) for HIITCYC.

For the GN no significant differences were found for condition ($p = 0.685$) or bout ($p = 0.057$) (Fig. 3A).

For the LVL, there was a main effect in the mean $\Delta$[HHb] for condition: [($p = 0.001$, $F = 17.647$, $\eta_p^2 = 0.616$) HIITCYC 10.74 ± 8.53 μM; HIITRUN 1.87 ± 3.43 μM]. There was no significant condition $\times$ bout interaction. For the mean $\Delta$[HHb] for each bout (Fig. 3B), differences were found between conditions for Bout 1 ($p = 0.002$, $F = 17.344$, $\eta_p^2 = 0.612$), Bout 2 ($p = 0.003$, $F = 14.593$, $\eta_p^2 = 0.57$), Bout 3 ($p = 0.001$, $F = 17.897$, $\eta_p^2 = 0.619$) and Bout 4 ($p = 0.001$, $F = 18.087$, $\eta_p^2 = 0.622$) with $\Delta$[HHb] higher during HIITCYC, compared to HIITRUN. For the mean $\Delta$[HHb] within conditions, a significant increase was found when comparing Bout 1–Bout 3 in the HIITCYC condition only ($p = 0.006$, $F = 7.994$, $\eta_p^2 = 0.421$).

For the RVL, there was a main effect in the mean $\Delta$[HHb] for condition: [($p = 0.002$, $F = 16.347$, $\eta_p^2 = 0.598$) HIITCYC 11.16 ± 7.99 μM; HIITRUN 3.51 ± 4.88 μM]. There was a condition $\times$ bout interaction ($p = 0.045$). For the mean $\Delta$[HHb] for each bout (Fig. 3C), differences were found between conditions for Bout 1 ($p = 0.002$, $F = 15.371$, $\eta_p^2 = 0.583$), Bout 2 ($p = 0.003$, $F = 13.762$, $\eta_p^2 = 0.556$), Bout 3 ($p = 0.002$, $F = 16.362$, $\eta_p^2 = 0.598$) and Bout 4 ($p = 0.001$, $F = 18.134$, $\eta_p^2 = 0.622$) with $\Delta$[HHb] higher during HIITCYC, compared to HIITRUN. For the mean $\Delta$[HHb] within conditions, there were no significant differences found across bouts.

### Systemic oxygen consumption

For the $VO_2$, there was a main effect for condition ($p = 0.017$, $F = 9.118$, $\eta_p^2 = 0.533$) and bout ($p = 0.008$, $F = 5.016$, $\eta_p^2 = 0.385$). There was a condition $\times$ bout interaction ($p = 0.021$, $F = 3.885$, $\eta_p^2 = 0.327$).

For the mean $VO_2$ for each bout (Fig. 4A), differences were found between conditions for Bout 2 ($p = 0.003$, $F = 14.807$, $\eta_p^2 = 0.597$) and Bout 3 ($p = 0.012$, $F = 9.317$,

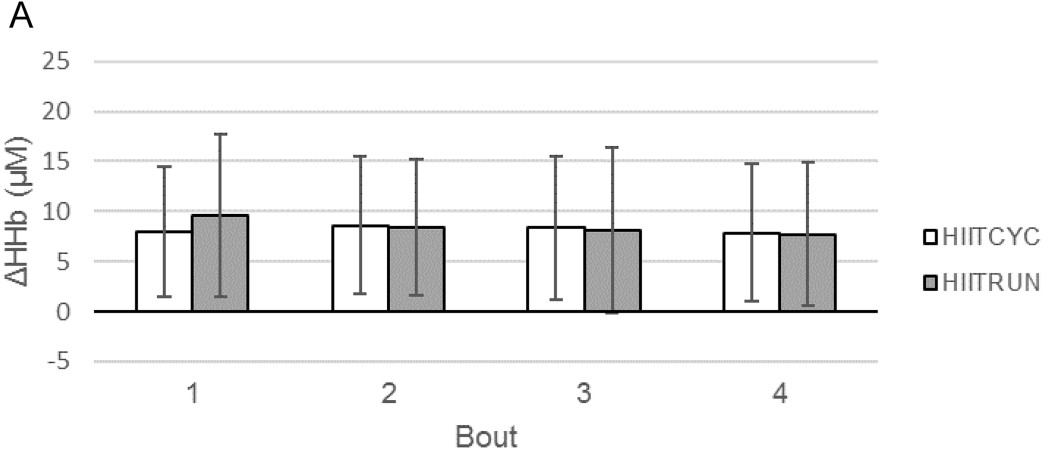

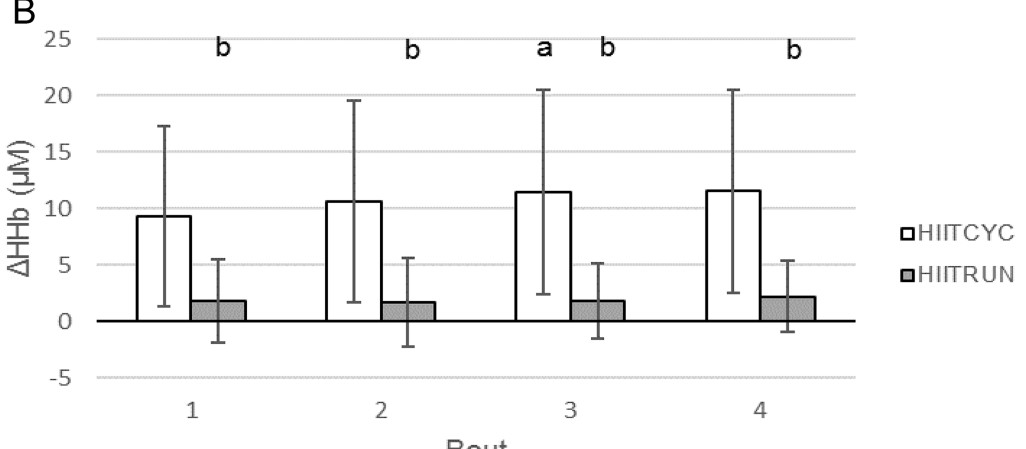

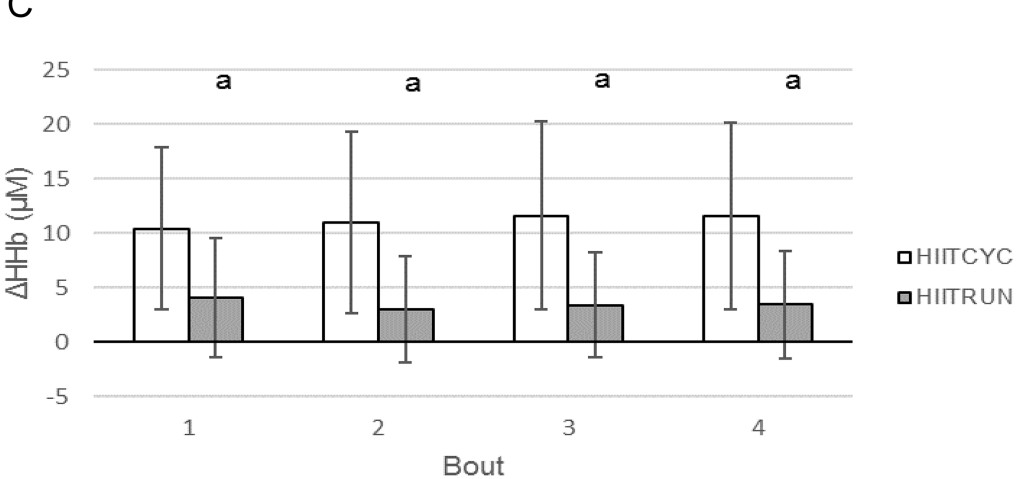

**Figure 3 Deoxygenated haemoglobin (HHb) concentration during the four bouts in cycling (HIITCYC) and running (HIITRUN) HIIT.** (A) GN. No significant differences between or within conditions. (B) LVL. a = significantly different to HIITCYC Bout 1; b = significantly different to HIITCYC during the same bout. (C) RVL. a = significantly different to HIITCYC during the same bout. Data are mean ± SD ($p \leq 0.05$).

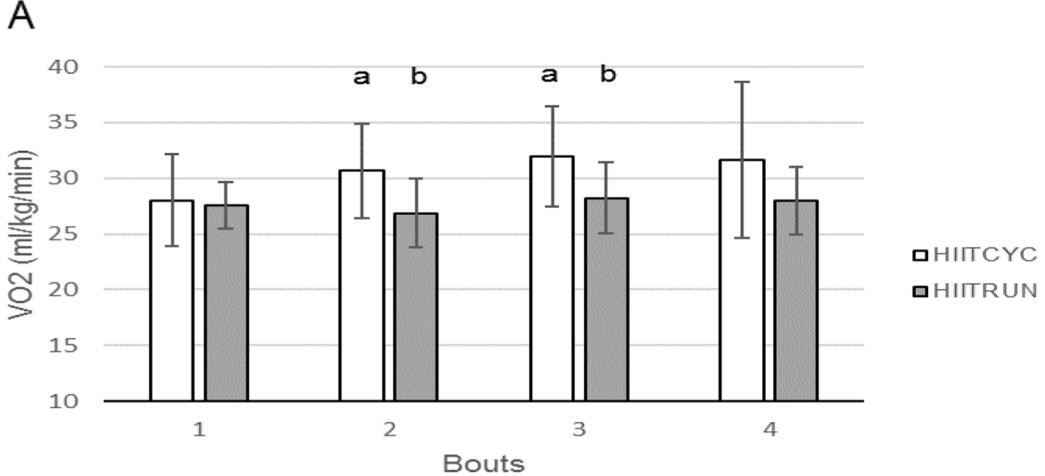

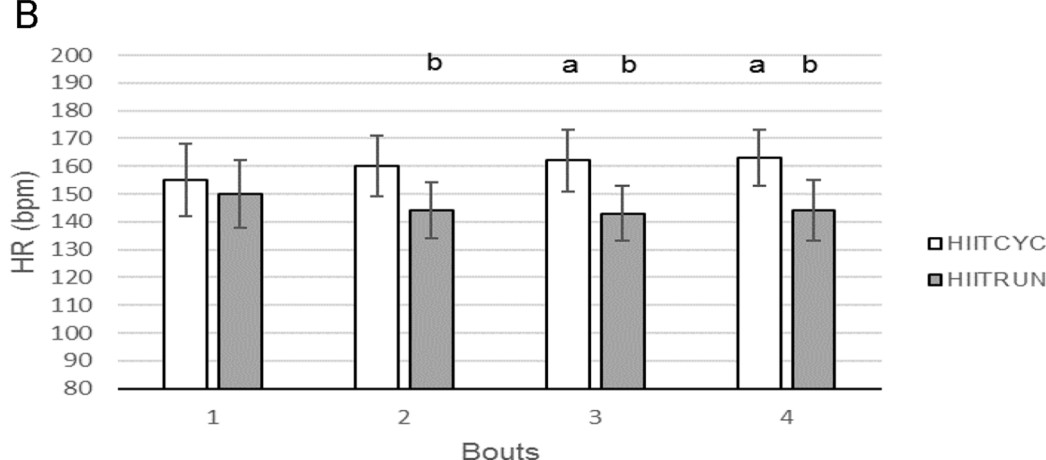

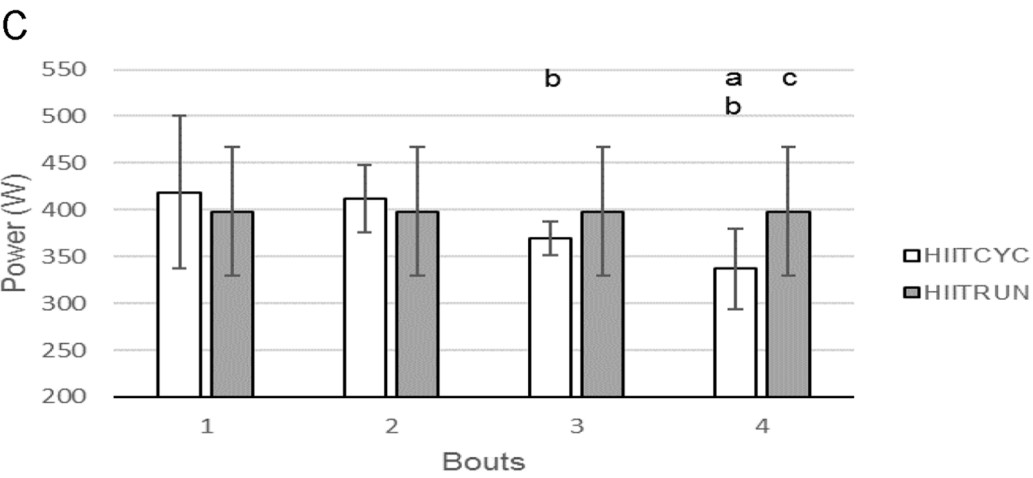

**Figure 4 Oxygen consumption, heart rate and mechanical power during the four bouts in cycling (HIITCYC) and running (HIITRUN) HIIT.** (A) VO2. a = significantly different to HIITCYC Bout 1; b = significantly different to HIITCYC during the same bout. (B) HR. a = significantly different to HIITCYC Bout 1; b = significantly different to HIITCYC during the same bout. (C) Mechanical power. a = significantly different to HIITCYC Bout 1; b = significantly different to HIITCYC Bout 2; c = significantly different to HIIITCYC during the same bout. Data are mean ± SD ($p \leq 0.05$).

$\eta_p^2 = 0.482$) with VO$_2$ higher during HIITCYC, compared to HIITRUN. For the mean VO$_2$ within conditions, values increased over time in HIITCYC ($p = 0.016$, $F = 6.347$, $\eta_p^2 = 0.414$) when comparing Bout 1 to Bout 2 and Bout 3.

### Heart rate

For the HR, there was a main effect for condition ($p < 0.001$, $F = 31.126$, $\eta_p^2 = 0.757$). There was a condition × bout interaction ($p = 0.014$).

For the mean HR for each bout (Fig. 4B), differences were found between conditions for Bout 2 ($p < 0.001$, $F = 26.002$, $\eta_p^2 = 0.722$), Bout 3 ($p < 0.001$ $F = 53.022$, $\eta_p^2 = 0.841$) and Bout 4 ($p < 0.001$, $F = 37.521$, $\eta_p^2 = 0.79$) with HR higher during HIITCYC, compared to HIITRUN. For the mean HR within conditions, values increased over time in HIITCYC ($p < 0.002$, $F = 12.316$, $\eta_p^2 = 0.804$) when comparing Bout 1 to Bout 3 and 4.

### Mechanical power

For the power output, there was a main effect for bout ($p = 0.001$, $F = 11.205$, $\eta_p^2 = 0.528$). There was a condition × bout interaction ($p = 0.001$).

For the mean power output for each bout (Fig. 4C), differences were found between conditions for Bout 4 ($p = 0.008$, $F = 10.728$, $\eta_p^2 = 0.518$) with mechanical power lower during HIITCYC, compared to HIITRUN. For the mean power output within conditions, values decreased over time in HIITCYC ($p = 0.001$, $F = 11.205$, $\eta_p^2 = 0.528$) when comparing Bout 1 and 2 to Bout 3 and 4.

### Ratings of perceived exertion

For the mean RPE, there was a main effect for condition ($p < 0.001$, $F = 79.976$, $\eta_p^2 = 0.885$) and bout ($p = 0.001$, $F = 14.259$, $\eta_p^2 = 0.765$). There was no significant condition × bout interaction.

For the mean RPE for each bout (Fig. 5A), differences were found for Bout 1 ($p = 0.001$, $F = 20.477$, $\eta_p^2 = 0.651$), Bout 2 ($p < 0.001$, $F = 95.703$, $\eta_p^2 = 0.905$), Bout 3 ($p < 0.001$, $F = 70.304$, $\eta_p^2 = 0.865$) and Bout 4 ($p < 0.001$, $F = 85.105$, $\eta_p^2 = 0.886$) with RPE higher during HIITCYC, compared to HIITRUN. For the mean RPE within conditions, values increased over time in the HIITCYC ($p < 0.001$, $F = 44.704$, $\eta_p^2 = 0.803$) and HIITRUN ($p < 0.017$, $F = 6.251$, $\eta_p^2 = 0.405$) conditions when comparing Bout 1 to Bout 2, 3 and 4 during HIITCYC and Bout 1, 2 and 3 to Bout 4 during HIITRUN.

The RPE$^{COND}$ score was higher for HIITCYC, compared to HIITRUN, ($p < 0.001$, $F = 37.027$, $\eta_p^2 = 0.771$) (Fig. 5B).
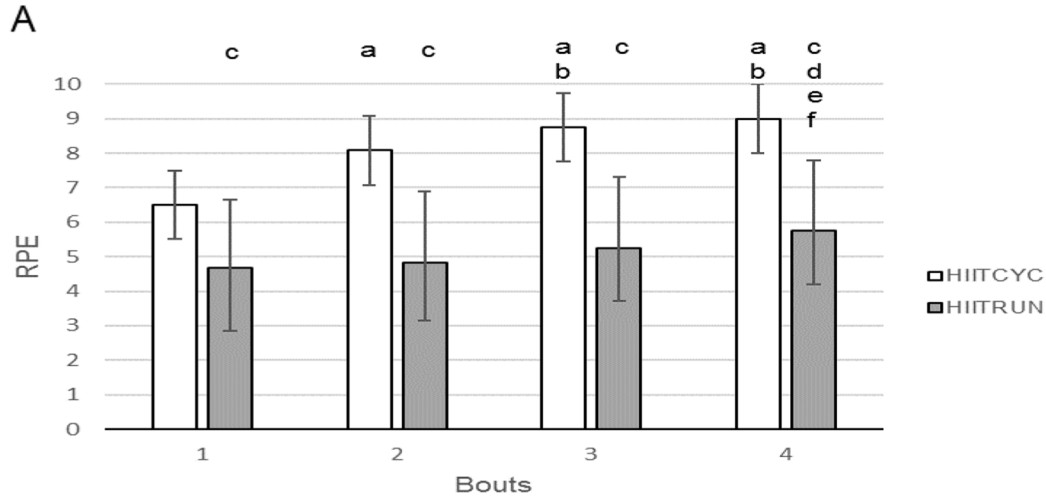

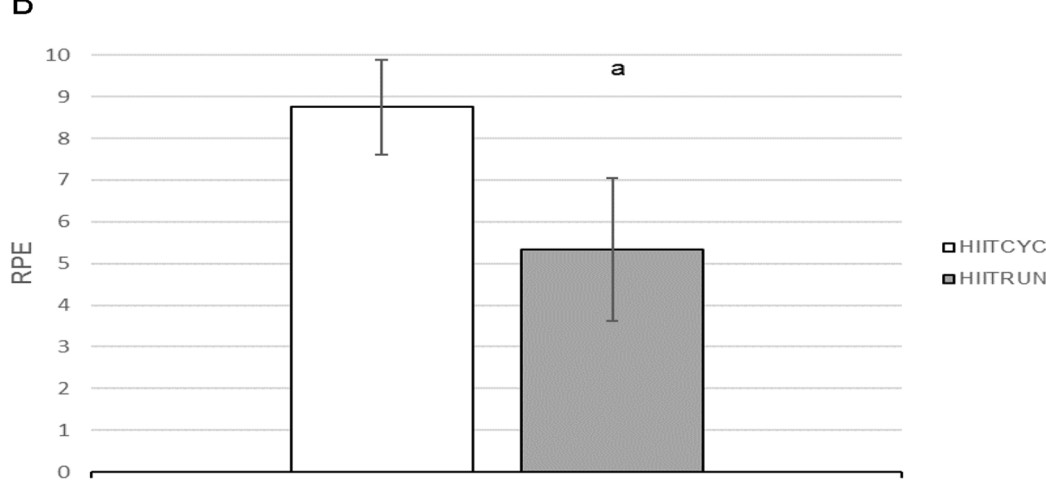

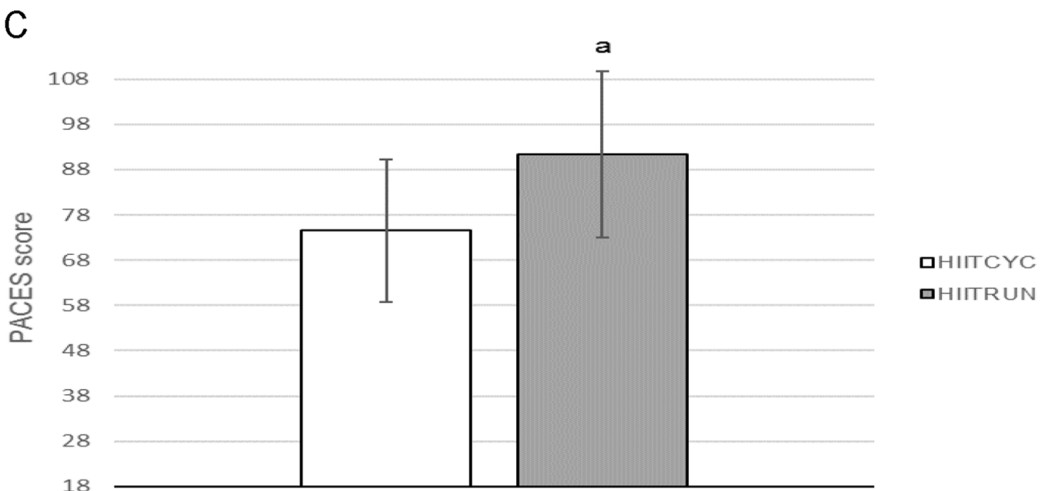

**Figure 5 Ratings of perceived exertion, condition ratings of perceived exertion and Physical Activity Enjoyment Scale for cycling (HIITCYC) and running (HIITRUN) HIIT.** (A) RPE. a = significantly different to HIITCYC Bout 1; b = significantly different to HIITCYC Bout 2; c = significantly different to HIIITCYC during the same bout; d = significantly different to HIITRUN Bout 1; e = significantly different to HIITRUN Bout 2; f = significantly different to HIITRUN Bout 3. (B) RPECOND. a = significantly different to HIITCYC. (C) PACES. a = significantly different to HIITCYC. Data are mean ± SD ($p \leq 0.05$).

## Physical activity enjoyment scale

The PACES score was found to be higher for HIITRUN when compared to HIITCYC; $t(11) = -2.460$, $p = 0.032$, $\eta_p^2 = 0.355$ (Fig. 5C).

## DISCUSSION

The aim of this study was to compare the local ($\Delta$[HHb]) and systemic (VO$_2$) oxygen utilisation, HR, RPE and enjoyment responses during HIIT protocols that exemplify the format of HIIT that sedentary individuals would perform utilising running and cycling ergometers in unsupervised recreational exercise settings.

In support of our hypotheses, RPE was lower and PACES was higher for the HIITRUN condition, compared to the HIITCYC condition. In contrast to our hypotheses, $\Delta$[HHb] at the LVL and RVL sites, VO$_2$, and HR were higher during HIITCYC, compared to HIITRUN. No significant differences were found in $\Delta$[HHb] at the GN site when comparing HIITCYC and HIITRUN. It is important to highlight that, while determining the potential physiological load and therefore potential health benefit of one mode over another will not in itself reduce sedentary behaviour, this information will enable sedentary individuals to select the mode of exercise that would potentially facilitate greater health benefits or improve adherence.

### Tissue oxygenation

A significantly higher $\Delta$[HHb] at the LVL and RVL sites occurred during all bouts of HIITCYC, compared to HIITRUN. This is indicative of increased local oxygen utilisation in the large locomotor muscles during HIITCYC which is consistent with a higher systemic oxygen utilisation, as indicated by higher VO$_2$ during HIITCYC. It is conceivable that in a population with no specificity of training in cycling, during high-intensity exercise, a higher local oxygen utilisation (linked to metabolic demand) will exist during HIITCYC, compared to HIITRUN. This is irrespective of the fact that during incremental and steady-state exercise vastus lateralis muscle activity has been found to be greater or equal during running, compared to cycling (*Bijker, de Groot & Hollander, 2002*; *Bouillon et al., 2016*; *Millet & Lepers, 2004*).

In the smaller GN muscle of the leg, which generally has a greater percentage of oxidative muscle fibres (*Hagström-Toft et al., 2002*; *Houmard et al., 1998*) and greater citrate synthase activity (*Houmard et al., 1998*) than the vastus lateralis muscle, there were no significant differences in $\Delta$[HHb] between conditions or bouts. In muscles comprised of oxidative fibres, an improved matching of oxygen supply and demand has been shown, when compared to muscles comprised of glycolytic fibres

(*Boone et al., 2016*). This indicates that during exercise, GN was able to meet the local increased energy requirements, creating little change in HHb values compared to pre-exercise values, irrespective of the mode specific differences in systemic and vastus lateralis oxygen utilisation. Additionally, while GN muscle activation is generally higher during cycling than running at the same HR (*Bouillon et al., 2016*), it has been shown that running on an incline leads to a greater activation of the GN than during level running (*Millet, Vleck & Bentley, 2009*). Therefore, it is conceivable that the greater activation of GN during the incline running protocol of HIITRUN resulted in a similar level of oxygen utilisation in GN as the cycling protocol during HIITCYC.

When comparing $\Delta$[HHb] within conditions, there was no overall progressive increase in muscle oxygen utilisation in GN, LVL and RVL from Bout 1 to Bout 4 irrespective of exercise mode. Exercise effort was submaximal (as indicated by $VO_2$, HR and RPE data) and if the 2 min passive recovery periods allowed adequate reoxygenation and recovery in the vasculature of the working muscles, the muscles would be able to meet the energy requirements of each bout with existing oxygen supply, creating little change in HHb values across bouts. The lack of significant change in $\Delta$[HHb] at the locomotor muscle sites from Bout 1 to 4 indicates that the level of maximal oxygen utilisation achieved during the conditions was not increased by the addition of multiple exercise bouts, irrespective of the mode of exercise adopted. This suggests that a single bout may be sufficient to induce maximal levels of oxygen utilisation, and this has implications for identifying the smallest dose of HIIT needed to convey the health benefits attributed to this format of exercise.

The inter-individual variability in the RVL $\Delta$[HHb] response during the HIITCYC condition, as illustrated in Fig. 2, was unrelated to the participants power output (i.e. the participants with the highest power outputs did not show the greatest increases in HHb). A similar level of variability in the $\Delta$[HHb] response can be seen in the two projects to publish individual results (*Jones, Hamilton & Cooper, 2015*; *Kriel et al., 2016*). While recommendations for performing muscle NIRS measurements and NIRS data analysis are available (*Ferrari, Muthalib & Quaresima, 2011*; *Grassi & Quaresima, 2016*; *Perrey & Ferrari, 2018*), there is no accepted standard for the method of calculating, analysing and presenting individual and group NIRS data. Furthermore, factors affecting the quality and integrity of NIRS data (such as ATT and optode positioning) are often not reported (*Perrey & Ferrari, 2018*). This makes comparison of $\Delta$[HHb] data between projects difficult, even when projects have been performed in similar populations, performing similar HIIT interventions.

To our knowledge, this is the first study to show that in sedentary participants, the $\Delta$[HHb] levels attained during HIIT varies in three locomotor muscles of the lower limbs, due to the mode of HIIT exercise. This finding indicates the specificity of oxygen utilisation in the sedentary population.

## Systemic oxygen consumption

Higher peak $VO_2$ and HR values are frequently reported during incremental running exercise, compared to incremental cycling exercise, due to the increased metabolic cost

of weight bearing ambulation (*Abrantes et al., 2012*; *Basset & Boulay, 2000*; *Hill, Halcomb & Stevens, 2003*; *Scott et al., 2006*).

In contrast, the $VO_2$ and HR responses were not different at Bout 1 between HIITCYC and HIITRUN, indicating that initially the conditions were well matched for intensity. Furthermore, significantly higher $VO_2$ during Bout 2 and Bout 3 of HIITCYC, compared to HIITRUN, indicates an increased exercise intensity and integrated physiological demand during the non-weight bearing cycling condition, contributed to by an increased local oxygen utilisation, as evidenced by the higher $\Delta$[HHb] discussed previously. In a sedentary population, it is assumed that at least a partial application of the specificity of training principle will exist for walking/running (due to everyday ambulation) while no such training effect would be present for cycling (no participants reported routine cycling activity when completing their physical activity log). This lack of adaptation, both centrally and peripherally, could have contributed to the increased intensity and physiological demand during HIITCYC.

When examining $VO_2$ changes within each condition, a potential explanation for there being no difference in $VO_2$ between Bout 2, 3 and 4 of HIITCYC could be the significant decrease in mechanical power output across bouts, effectively requiring less aerobic contribution over time, in effect counteracting the expected continued increase in $VO_2$ in later bouts due to the cumulative load of the protocol. Exercise effort was submaximal during HIITRUN (as indicated by $VO_2$, HR and RPE data) and it is conceivable that the 2 min passive recovery periods allowed adequate recovery from this submaximal exercise. Therefore, $VO_2$ remained relatively stable over time.

The sedentary participants were potentially limited in their ability to achieve true maximal running speeds during the incremental treadmill test (MAX) due to possible slower $VO_2$ kinetics, poor running efficiency, and lack of familiarity with running at rapidly increasing speeds at a 10% gradient. This is however an important observation: it appears that the (in)ability to run at speeds associated with maximal/supramaximal exercise may limit the exercise intensity able to be achieved by sedentary individuals during short duration treadmill running HIIT when compared to short duration cycling HIIT. A potential future research direction may be to compare the beneficial physiological and perceptual effects of less time-efficient, but submaximal HIIT protocols performed on the treadmill, such as the $4 \times 4$ min protocol popular in clinical research (*Ramos et al., 2016*), to Wingate cycling HIIT in sedentary participants.

### Heart rate

The significantly higher mean HR during Bout 2, 3 and 4 of HIITCYC, compared to HIITRUN, are further indications of an increased exercise intensity and physiological demand during the cycling condition.

The relatively static HR response over time, when comparing Bout 2, 3 and 4 of HIITCYC, supports similar findings in which a passive recovery HIIT protocol was used (*Lopez, Smoliga & Zavorsky, 2014*). In the HIITRUN condition, similar to $VO_2$ values, exercise effort was submaximal and the 2 min recovery periods allowed adequate recovery, it is conceivable that HRs would also remain relatively stable over time.

## Mechanical power

The most frequently used modes of HIIT are running and cycling, utilising ergometers (*Kessler, Sisson & Short, 2012*; *Logan et al., 2014*; *Weston et al., 2014*). However, as mentioned previously, there are differences in how running and cycling ergometers are utilised to induce the requisite physiological stress. Cycle ergometer HIIT, during which participants do not support their own body weight, safely lends itself to all-out efforts, and this is typically achieved (*Weston et al., 2014*). During treadmill ergometer HIIT, utilising motorised treadmills, the ability to safely produce short duration all-out efforts is limited. Typically, a prior indication of maximal running speed is required so that the speed (and gradient) of the HIIT session can be set to produce a high-intensity effort (*Ben Abderrahman et al., 2013*; *Iaia et al., 2009*). Therefore, the mechanical power produced during HIITRUN and HIITCYC were not directly matched. However, when comparing mechanical power between conditions, no difference in mechanical power output occurred until Bout 4, when the mechanical power in HIITCYC was lower than in HIITRUN. The late reduction in mechanical power during HIITCYC, compared to HIITRUN, could be attributed to a greater cumulative fatigue due to the all-out format of the cycling bouts.

Power output (and exercise intensity) remained stable and submaximal for HIITRUN from Bout 1 to Bout 4. The submaximal intensity potentially enabled a more consistent effort during exercise, and enhanced recovery, during HIITRUN, hence potentially less cumulative fatigue developed. The higher relative intensity of the all-out cycling bouts during HIITCYC potentially lead to the significant decline in power output from Bout 1 to Bout 4 and the higher $\Delta$[HHb], $VO_2$ and HR responses (*Dupont et al., 2007*; *Lopez, Smoliga & Zavorsky, 2014*; *Wahl et al., 2013*). This is further evidence of cumulative fatigue during the cycling condition and was expected due to the incomplete ATP repletion and phosphocreatine resynthesis that occur during repeat Wingate exercise that do not allow time for complete recovery (*Dupont et al., 2007*; *Lopez, Smoliga & Zavorsky, 2014*; *Wahl et al., 2013*). The differences in mechanical power output and relative intensity between HIITRUN and HIITCYC is an important finding, demonstrating that when utilising typical HIIT protocols available to sedentary individuals looking to initiate physical activity in recreational exercise settings, cycling is likely to result in greater perturbations in physiological responses compared to running.

While the aim of this study was not to match the exercise intensity of running and cycling HIIT sessions, future research in which absolute workload of the HIIT session is matched would assist in determining to what extent the differences in responses observed in this study were the result of mode and/or intensity.

## Ratings of perceived exertion

Ratings of perceived exertion was higher during the first bout of HIITCYC, compared to HIITRUN, despite no difference in markers of physiological intensity ($VO_2$ and HR) or mechanical power. This finding is consistent with previous research indicating that exercise performed at the same absolute intensity corresponds to a higher relative intensity in cycling (*Abrantes et al., 2012*). RPE was also higher during Bouts 2–4 of

HIITCYC compared to HIITRUN. This finding is consistent with previous research which incorporated incremental (*Abrantes et al., 2012*) or steady state (*Thomas et al., 1995*) exercise. RPE is a subjective indicator of intensity during exercise (*Pescatello & American College of Sports Medicine, 2013*) and taken together with the higher $\Delta$[HHb], $VO_2$ and HR response during Bouts 2–4 of HIITCYC provide further evidence of a higher physiological strain, a higher level of cumulative fatigue during HIITCYC and a potential lack of habitual adaptation to cycling.

## Physical activity enjoyment scale

The examination of sedentary individuals' enjoyment during exercise has provided inconsistent results: Sedentary individuals are more likely to enjoy and be compliant with exercise that they perceive as moderate in intensity (*Williams et al., 2008*), however HIIT has also been shown to be enjoyable and elicit a degree of positive affect (*Martinez et al., 2015*). Exercise mode has been shown to have an acute effect on psychological mood state, including enjoyment, with running eliciting a more positive mood profile than weightlifting (*Dyer & Crouch, 1988*). However, the effect of mode on sedentary individuals' enjoyment levels during HIIT is unknown.

Participants enjoyed the HIITRUN session more. When taking into account the significantly lower RPE scores in HIITRUN when compared to HIITCYC, this finding is in agreement with research that has found that exercise that elicits a lower RPE score is more enjoyable (*Kilpatrick et al., 2003*).

Adverse reactions to HIIT exercise are rarely reported (*Weston, Wisloff & Coombes, 2014*). During the current study, seven participants reported either leg muscle discomfort/cramping ($n = 3$) or dizziness and nausea ($n = 4$) during the recovery period of the HIITCYC condition. Although all participants felt well upon leaving the laboratory (no more than 30 min post exercise), these physiological responses could have contributed to the lower levels of enjoyment during HIITCYC.

## Limitations

Up to nine design variables can be adjusted when designing HIIT protocols (*Buchheit & Laursen, 2013b*), leading to large variability in HIIT protocol composition within the literature. The 30 s bouts and 2 min recovery periods (a 1:4 work to recovery ratio) limits the generalisability and comparison of the research findings to the broader HIIT literature.

It is routine to wait ≈15 min to collect an 'overall RPE' after HIIT, due to the discomfort of the last bout potentially affecting the overall figure. However, adverse reactions experienced by seven participants during this study were noted in previous studies and generally occurred during the latter half of the 6 min post-exercise recovery period. Collecting the RPE$^{COND}$ score earlier in recovery, while potentially introducing an exercise related bias and therefore a potential limitation to this study, avoided a potential bias due to adverse sensations experienced later in recovery.

The PACES data were collected retrospectively, following exercise. The PACES score was likely to be influenced by the participants' physiological and psychological state at

that time. This study would perhaps have been improved by assessing enjoyment during exercise, however this was deemed impractical due to experimental design and physiological data collection methods.

It is acknowledged that sedentary participants can exhibit slower $VO_2$ kinetics, hence $VO_{2max}$ values may have been underestimated by using stages of 30 s duration during the maximal incremental exercise test, compared to standard stages of 3 min duration. However, no difference in $VO_{2max}$ has been shown with short versus traditional length protocols (*Kang et al., 2001*; *Midgley et al., 2008*). Furthermore, a $VO_{2max}$ of $43.5 \pm 4.3$ ml $\cdot$ kg $\cdot$ min$^{-1}$, classified as the 35th percentile in terms of fitness (*American College of Sports Medicine et al., 2018*; *Kaminsky, Arena & Myers, 2015*), is relatively high for sedentary participants, making underestimation of $VO_{2max}$ unlikely given that the reported mean physical activity time of a moderate intensity for this group was $31 \pm 33$ min $\cdot$ week$^{-1}$.

## CONCLUSIONS

It is concluded that, in sedentary individuals, free-paced cycling HIIT produces higher levels of physiological stress when compared to constant-paced running HIIT. A single bout of HIIT may be sufficient to induce maximal levels of muscle oxygen utilisation, irrespective of the mode of exercise. Participants perceived running HIIT to be more enjoyable than cycling HIIT. These findings have implications for selection of mode of HIIT for physical stress, exercise enjoyment and compliance.

## ACKNOWLEDGEMENTS

The authors would like to thank the participants of this research, without whom this study would not have been possible. Thank you to Dr. Hugo Kerhervé for his technical assistance.

### Funding
This project was supported by an annual research student allocation. The funders had no role in study design, data collection and analysis, decision to publish, or preparation of the manuscript.

### Competing Interest
The authors declare that they have no competing interests.

### Author Contributions
- Yuri Kriel conceived and designed the experiments, performed the experiments, analysed the data, prepared figures and/or table, authored or reviewed drafts of the paper, approved the final draft.
- Christopher D. Askew analysed the data, authored or reviewed drafts of the paper, approved the final draft.

- Colin Solomon conceived and designed the experiments, analysed the data, authored or reviewed drafts of the paper, approved the final draft.

## Human Ethics

The following information was supplied relating to ethical approvals (i.e., approving body and any reference numbers):

This research project was approved by the Human Research Ethics Committee of the University of the Sunshine Coast (S/13/472).

## Data Availability

Kriel, Yuri (2017): Yuri Kriel - article data: Mode. figshare. Dataset. https://doi.org/10.6084/m9.figshare.5271001.v1

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
