# Peer review of "The effect of running versus cycling high-intensity intermittent exercise on local tissue oxygenation and perceived enjoyment in 18–30-year-old sedentary men"

_PeerJ, doi:10.7717/peerj.5026_

## Round 0.1 · original submission · Major Revisions

The reviewers have identified several severe issues with the manuscript that need to be addressed by the authors including but not limited to workload differences between groups. If the authors can effectively address these severe issues as well as the other issues raised by reviewers, please provide a point by point response to each concern raised and how and where in the revised manuscript the issue was addressed.

Reviewer 1 ·

Basic reporting

This manuscript is very well written, with only a couple of minor grammatical oversights.
The literature and references are primary and pertinent and sufficient background is provided to establish context.

The results shown answer the hypothesis provided.

Experimental design

This work is original. It performs incremental work within the areas of HIIT, NIRS and exercise prescription.
The research question is well defined, certainly relevant. If exercise adherence and health benefits within sedentary populations can be obtained through time efficient and enjoyable exercise modalities, this is then of obvious benefit.
The authors should perhaps mention that within the introduction/discussion that establishing the potential health benefit of one mode of exercise over another is not the answer to removing sedentary behaviour nor ensuring exercise adherence. It does however allow the practionner to direct an individual to a potentially more favourable exercise session- that may aid adherence, greater facilitate health benefits etc.
The experiment has been carried out with rigour, although I must admit to not fully understanding the rationale of not matching the workloads. If these are not matched, then comparisons between the two modes of exercise on physiological strain and self-report measures appears to be irrelevant, as clear differences will be present. The authors discuss matched workloads (submax/incremental) (lines56-59) within the introduction and state that this work has not been carried out with sedentary populations and so I wonder why this was not carried out as the experiment. I in no way mean to be obtuse.
The authors explain that as the workload between running and cycling is likely to differ it is important to describe this to better explain which modality is more ‘beneficial’ for sedentary individuals. This work therefore looks to detail the differences between different modes and intensities of exercises and therefore will find inherent differences. I again do not mean to appear dismissive, I just question the meaningfulness and perhaps this needs to be more explicitly explained.

Validity of the findings

Figure 1 is clear and well presented.
Figure 2 panel A may benefit from a scaling change to make the data more obvious
Figure 3 is clear and well presented.
Figure 4 is clear and well presented
Table 1 is clear and well presented
Raw data is suggested within a table or written within the results text for figure 3 as large SD's are shown for panel C (Power). It is important to see this data as whilst the authors point out bike and run 'HIIT' sessions were not matched, nor meant to be; the closer the matching of the work intensities the more relevant this work is. Further raw data would benefit the NIRS data with individual data shown. The concept of individual prescription is applicable in elite athletes and sedentary populations for exercise adaptation.
The conclusion is structured logically, discusses the findings in detail and is linked back to the original research question.
I do not agree that either protocol can be called a HIIT protocol. This is largely based upon Figure 3 panel B; HR values would need to be c.180bpm and Figure 4 panel B; RPE scores for HIIT RUN are also not of the required level for maximal to near maximal efforts. It is suggested that HIIT be changed for submaximal efforts, unless the data does support a HIIT protocol.
Lines 387-391. I am not sure if the lower metabolic cost of eccentric muscle actions can be used to as an explanation for the lower HHb findings in the RUN exercise as RUN exercise has both eccentric and concentric actions. Perhaps the explanation should be removed. Lines 380-387 provide ample explanation.

I must admit again that I do not fully understand the rationale of the research. The authors state in lines 554-555 the perceptual responses to the two modes of exercise which essentially that; Exercising at a lower physiological stress, is reported as easier by the participants.
I welcome the authors to explain the importance and make this clear and obvious to the readers, so they like myself largely question why the workload was not matched.

Additional comments

I must commend the authors for this written work. It is well presented and has been carried out well.
I do not mean to dismiss the rationale behind the work, I just do not fully understand the premise or not work matching the cycle and run protocols.
I do feel that data individualisation will benefit the work and show the individual responses to both modalities, as with all exercise studies valuable data can be lost with the means.

Separate point:
Lines 209-212: I am interested in the reported movement artefact during the rest periods. This is unusual. Did this occur in the TSI% signal as well?

Minor grammatical:
Line 181: ofHITT ,
Line 202: previously(
Line 391: HIITRUN (full stop missing)

Reviewer 2 ·

Basic reporting

Scientific bakground for this study is insufficient.
Structure of the manuscript should be reviewed in depth.
Please see General comments.

Experimental design

The introduction needs more detail and to be targeted on the main factors.
Some parts of the experimental design are not related to the research question.
Experimental design as proposed appears incomplete.
Please see General comments.

Validity of the findings

Data cannot be judged and discussed properly based on the experimental design used.
Please see General comments.

Additional comments

General comments

Authors want to study the effect of 2 modes of dynamic whole body exercises (running versus cycling) during high intensity intermittent exercise on various physiological variables and perceived enjoyment in sedentary adults. This topic is of interest. A high intensity interval training (HIIT) exercise format was proposed in a single session.

As currently proposed, HIIT is not really defined and characterized throughout the manuscript. Authors should refer to a stronger scientific background on HIIT. What is lacking is any consideration on “high intensity” beside the low volume for HIIT. HIIT is an enhanced form of interval training involving brief, high-intensity, anaerobic exercise (ranging from 85% to 250% VO2 max for 6 s to 4 min) separated by brief, but slightly longer bouts of low-intensity aerobic rest (ranging from 20% to 40% VO2 max for 10 s to 5 min).
With regards to the methods and results sections, less information on exercise intensity is proposed. We can only extract from Figure 3 (and Table 1) that intensity level during the two exercises is around 60-65%VO2 max, <80%HRmax with a mean power output of 400W. Please adjust accordingly. Finally one fundamental question is to know what HIIT format / model was really used in the present study? How it was used ?
Two modes of exercise were compared. However, the two modes seem not matched according to some points highlighted below.
First, in their experimental design, authors used free-paced cycling compared to constant-paced running HIIT. However, constant- and free- paced movement produce likely different levels of physiological stress without considering the mode of exercise (running vs. cycling). Thus, mode of pacing is one confounding factor. Authors have to comment on this choice that represents a serious issue.
Second we do not know the workload that was manipulated for both modes of exercise (it only stated during the warm up phase). In addition to control the movement pace, similar intensity in terms of absolute workload and relative workload (based on % of VO2max for a specific exercise mode) are needed for comparison purpose. Methods are poorly detailed on intensity for the two modes. Only based on Figure 3, we can try to judge what were the intensity levels. See previous comments on that issue.

The Introduction and Discussions sections are too long. I would ask the Authors to re-arrange them in order to make the manuscript easier to read and understand.

In the Introduction, there are no real rationale on the use of running or cycling in terms of physiological / perceived responses to single HIIT session. Authors have to inform readers that the two modes of whole body exercise are based on different muscle action patterns (concentric for cycling and eccentric/concentric balance for running according to the slope; uphill running being more concentric, see reference of Minetti et al.). Importantly, information is lacking to understand why authors proposed this hypothesis. What are the relevant parameters that readers should know for such expectation on running? This is unclear.
In the Methods, authors proposed to investigate several muscle sites for evaluating oxygenation patterns but nothing appears in the introduction on this second independent factor. Again, what we know or not on regional muscle oxygenation profile for running and cycling deserves information and comments? What are the possible expectation on this point according to the 2 modes of muscle exercise? This is again unclear. All text on prefrontal cortex oxygenation is out of scope (as for fatigue ..) and should be removed. Please focus only on muscle oxygenation patterns during HIIT for 2 modes of whole body exercise. The last independent factor (time or repetition) needs also to be introduced.
To sump up, introduction, as currently proposed, is incomplete and not focused.

The description of the methodologies should report more details about the different methods used for estimating exercise intensity during the HIIT protocol used for running and cycling. In the current version, most of the details are reported by means of references to previous or other works. I would suggest to insert some formulas to allow the reader to directly comprehend the steps of the protocols and analysis.

Specific comments

Introduction
Page 7, first paragraph. It is important to define in a complete way what is HIIT and to inform clearly on what we know on the effects of a single HIIT session in running / cycling. Authors just reported “these investigations examined… have been evaluated previously etc..” but no findings are highlighted.
Page 7, second paragraph, L 56. What means “greater cardiorespiratory..” please inform readers. “Matched workload” has to be precised: absolute, relative, and domains of intensity?
Page 8, sentence 64-70 needs to be rephrased.
Page 8, line 75. What does mean “component”. Please explain.
Page 8, lines 75-77. What we have to retain from these references?
Page 8, HHb is not giving the same information on muscle as compared to cerebral tissue and cannot be used as currently proposed. As said before, only muscle oxygenation profiles have to be studied.
Page 8, Lines 83-84.. This needs to be argued. Please do so.
Page 9, line 86. The sentence like “local tissue oxygen utilisation has not been compared across exercise modes in three distinct locomotor muscles and the pre-frontal cortex” cannot justify a scientific study!
Please try to give information on the primer muscles used for each mode to explain the possible differences in muscle oxygenation profile according to the muscle sites.
Page 9, lines 87-91. This sentence needs to be argued. Please use adequate and relevant references for that purpose.
Page 10, line 110. The rationale based on scientific background on the possible differences between running and cycling for pulmonary oxygen consumption is absent.

Methods
Line 126. A protocol was done in constant pace while the other protocol was done in free pace. This is an important issue that could flaw the results.
Lines 163-168. This paragraph needs to be completed. See major comments.
Lines 172-175. The validity of this approach has to be commented.
Line 179. An incremental cycling test to exhaustion in order to get specific VO2max to the mode of exercise is needed in the design.
Lines 180+. What were the intensities of the HITT format for running and cycling exercises?
Line 189. Replace by “muscle oxygenation” here and thereafter.
Remove all information on prefrontal cortex.
Line 197. Please report also the ICC values for the reliability testing.
Line 210. Please explain this sentence and especially the values of SNR “during the passive recovery periods of the HIIT sessions, attributed to a low signal to noise ratio.” This is unclear.

Line 267. Results. For all statistical tests, report F values then p values and effect size estimates.
As stated before, since there is no rationale on the 3 factors (only 1 was partially introduced: mode of exercise) used in the introduction, this paragraph should be reviewed accordingly.

Figures: please, make them clearer (most of the panels are hard to understand, and they should be separated in different figures) and with higher quality. Also, some of the results could be transformed in statistical graphs rather than reporting all. Text and figures are redundant.

Several results have been reported throughout the manuscript. However, most of them are not statistically significant. I would suggest to report only the significant ones to emphasize the evidences, and make the manuscript easier to read.
The Discussion section is too long and should be re-arranged in order to report only the main findings and wrap up the impact of the work.

Note that there are some format errors (for reference and punctuation) throughout the manuscript.

·

Basic reporting

Well written and organized.
The introduction and discussion are rather lengthy and could be consolidated.

Experimental design

I appreciate the work undertaken by the authors - it is an important and relevant area of study. Moreover, the effect of HIIT modality on physiological and enjoyment responses is a novel area of study. The authors are to be commended.

In general, the experimental design is robust.
However, I have two primary concerns for the authors to address in their revision:
1. Was an apriori power calculation performed to identify necessary sample size?
2. It appears based on Figure 3C that participants were overly ambitious on the first bout of HIITCYC and consequently experienced a sustained decreased in power output for the remaining trials. Not surprisingly, physiological responses were higher (maybe not significantly but meaningfully yes) and also RPE. Accordingly, and again not surprisingly, PACES scores for HIITCYC were lower. In contrast, power outputs were sustained across bouts 1-4 for HIITRUN with associated lower physiological responses and RPE values coupled with higher PACES scores. My sense is that if power output across bouts of HIITCYC were consistent with those for HIITRUN that the findings would have been entirely different.

Validity of the findings

Per my last comment in the experimental design section - I don't question the validity of the findings based on the existing workload performance differences between HIITCYC and HIITRUN across bouts 1-4. However, had these been controlled and not existed the findings would have been much different. From an ecological validity standpoint and my own clinical experience with HIIT - it would be highly unusual to let previously sedentary individuals self-select their own HIIT intensity for the exact reasons observed in this study. If an individual with limited experience on a modality (i.e., cycle ergometer) is permitted to self-select the intensity, the exact scenario that occurred in this study is predictable. Individuals over-extend themselves early and become overly fatigued by the conclusion of the session (and don't have a positive response). A further problem is the high number of adverse responses experienced by the HIITCYC with the free-paced approach.

In summary, I suggest the authors address this fundamental limitation to their experimental design and findings and revise accordingly whilst taking these into account.

Additional comments

No additional comments.

---

## Round 0.2 · Minor Revisions

Although the manuscript was substantially improved, a few additional concerns remain. Please address these issues and provide a point by point response regarding how the issues were addressed.

Reviewer 2 ·

Basic reporting

Article is well structured as a whole. However at some places, the length of the manuscript can be lowered.

Experimental design

Clarity for the experimental design remains an issue (see below).

Validity of the findings

Data is robust. See below one comment on the use for the three factor, repeated-measures analysis of variance.

Additional comments

Thanks to the authors for this well revised manuscript that now clearly stated that the study aims to determine the physiological load between two “self-selected” modes of HIIT exercise (running vs. cycling). Authors provided adapted responses to the first comments and argued for some methodological choices. Initially, for the methods section, it was requested to inform more as a whole. Sorry for some misunderstandings in previous comments.

A few more comments need however to be taken account:

Abstract
L 23. The “efficacy of this format exercise” cannot be evaluated with the experimental design of the study. Change accordingly.

L 25. Since this study did not match the intensity in absolute and relative objective mechanical / physiological indices (=dependant variable) between the two modes of exercise, authors have to state here and throughout the manuscript the “self-selected pace strategy” they have used.
If authors included in their study “bouts of Wingate cycling exercise and bouts of maximal speed running exercise” (as reported in their responses), please give this information for clarity purpose. But in this case, maximal speed running exercise is not a constant-paced running task as underlined? This is still confusing.

Introduction
L 100-104. This sentence is not useful for building the rationale of the study. Only discussion could use such perspective.

L 131-135
“The HHb in the pre-frontal cortex is increased during high intensity cycling exercise (Kriel et al. 2016; Smith & Billaut 2010), compared to pre-exercise values and could be a potential mechanism contributing to exercise cessation (Shibuya et al. 2004) and therefore influence the physiological responses to HIIT exercise.”

HHb is not the good signal of interest at the cerebral level to highlight. Local oxygenation mechanisms at the muscle are clearly different from those at the cerebral level. More important, since the present descriptive study does not aim to determine “potential mechanism influencing the physiological responses to HITT exercise”, all parts on frontal oxygenation is out of scope. Authors have to target on physiological (and perceptual) responses at the systemic and muscle levels. In addition, length of the manuscript needs to be lowered.

Methods
L 252; “30 s bouts of HIIT”; once again just report here the requested intensity level: “all-out” “self-paced”… See comments below and above on this issue.
L 269. Based on previous comment, PFC should be removed.

L 335. Scientific background in the introduction is well emphasised on the effect of condition (exercise mode). However, at this stage, we do not know why authors want to look at the influence of bout and site (included in the 3-ANOVA) on the dependant variables. No background (goal and hypotheses) was proposed on these two factors. Please clarify accordingly where needed.

Results.
L 345. Only one paragraph on Tissue oxygenation is enough; it is not necessary to write for each muscle one dedicated subsection.
L 356. Based on previous comment, this paragraph has to be removed.
Discussion
Comparison with other studies needs to report when needed how the level of workload was matched (subjective or objective) between modes of exercise. Otherwise comparisons cannot be done.

L 538-540. This is highly speculative.
L 541-545 : L 566-570. To remove
L 563. Please refer to the recent review on this topic (doi: 10.1007/s40279-017-0820-1)

L 689. The statement on “matched intensity running and cycling HIIT” is wrong.
How free-paced cycling HIIT was matched to constant-paced running HIIT?

·

Basic reporting

Basic reporting satisfies all criteria.

Experimental design

My previous comments on experimental design have been sufficiently addressed by the authors in their revised paper.

Validity of the findings

My previous comments on validity of the findings have been sufficiently addressed by the authors in their revised paper.

Additional comments

I have no further comments.
I appreciate the work the authors committed to revising their paper per my previous remarks.
This paper makes a strong contribution to the literature.
The authors are to be commended.

---

## Round 0.3 · accepted · Accept

All previous issues raised have been sufficiently addressed by the authors in their revised paper.

# Reviewer 2 ·

Basic reporting

My previous comments on this point have been sufficiently addressed by the authors.

Experimental design

no comment

Validity of the findings

My previous comments on validity of the findings have been sufficiently addressed by the authors in their revised paper.

Additional comments

I have no further comments. I appreciate the revised manuscript based on previous comments.